# The Extent to Which Artificial Intelligence Can Help Fulfill Metastatic Breast Cancer Patient Healthcare Needs: A Mixed-Methods Study

**DOI:** 10.3390/curroncol32030145

**Published:** 2025-03-02

**Authors:** Yvonne W. Leung, Jeremiah So, Avneet Sidhu, Veenaajaa Asokan, Mathew Gancarz, Vishrut Bharatkumar Gajjar, Ankita Patel, Janice M. Li, Denis Kwok, Michelle B. Nadler, Danielle Cuthbert, Philippe L. Benard, Vikaash Kumar, Terry Cheng, Janet Papadakos, Tina Papadakos, Tran Truong, Mike Lovas, Jiahui Wong

**Affiliations:** 1Department of Psychiatry, University of Toronto, Toronto, ON M5S 1A1, Canada; 2College of Professional Studies, Northeastern University, Toronto, ON M5X 1E2, Canada; 3de Souza Institute, University Health Network, Toronto, ON M5G 2C4, Canada; 4Department of Medical Oncology, Princess Margaret Cancer Centre, University Health Network, Toronto, ON M5G 2C4, Canada; 5Department of Supportive Care, Princess Margaret Cancer Centre, University Health Network, Toronto, ON M5G 2C4, Canada; 6Cancer Education Program, Princess Margaret Cancer Centre, Cancer Health Literacy Research Centre, Toronto, ON M5G 2M9, Canada; 7Techna Institute, University Health Network, Toronto, ON M5G 2C4, Canada; 8Design and Innovation at Cancer Digital Intelligence, University Health Network, Toronto, ON M5G 2C4, Canada

**Keywords:** metastatic breast cancer, AI-assisted healthcare, patient education, chatbot

## Abstract

The Artificial Intelligence Patient Librarian (AIPL) was designed to meet the psychosocial and supportive care needs of Metastatic Breast Cancer (MBC) patients with HR+/HER2− subtypes. AIPL provides conversational patient education, answers user questions, and offers tailored online resource recommendations. This study, conducted in three phases, assessed AIPL’s impact on patients’ ability to manage their advanced disease. In Phase 1, educational content was adapted for chatbot delivery, and over 100 credible online resources were annotated using a Convolutional Neural Network (CNN) to drive recommendations. Phase 2 involved 42 participants who completed pre- and post-surveys after using AIPL for two weeks. The surveys measured patient activation using the Patient Activation Measure (PAM) tool and evaluated user experience with the System Usability Scale (SUS). Phase 3 included focus groups to explore user experiences in depth. Of the 42 participants, 36 completed the study, with 10 participating in focus groups. Most participants were aged 40–64. PAM scores showed no significant differences between pre-survey (mean = 59.33, SD = 5.19) and post-survey (mean = 59.22, SD = 6.16), while SUS scores indicated good usability. Thematic analysis revealed four key themes: AIPL offers basic wellness and health guidance, provides limited support for managing relationships, offers limited condition-specific medical information, and is unable to offer hope to patients. Despite showing no impact on the PAM, possibly due to high baseline activation, AIPL demonstrated good usability and met basic information needs, particularly for newly diagnosed MBC patients. Future iterations will incorporate a large language model (LLM) to provide more comprehensive and personalized assistance.

## 1. Introduction

Metastatic Breast Cancer (MBC) occurs when cancer cells spread to distant organs such as bones, the brain, lungs, liver, or other parts of the body, necessitating delivery of sequential systemic therapies to prolong life and manage symptoms until the end of life. However, navigating symptom management and treatment options can be complex for patients. Research highlights that MBC patients did not receive sufficient personalized information or emotional support due to a lack of MBC knowledge among allied health professionals [1]. Patients commonly expressed disease information needs, including symptoms and side-effect management, prognosis, and treatment options that can extend survival [1,2,3]. Patients experienced anxiety and depression due to critical and life-altering changes in physical appearance, employment, social activities, and family relationships [1,3,4]. Patients were also worried about death and dying, particularly the fear of pain and the sense of uncertainty [4]. Patients were often worried about their family’s future, including financial stability and adaptation after bereavement [5,6]. Nevertheless, survival was a primary concern for most MBC patients [4], with clinical trials and treatment options targeting the tumour’s genetic and molecular characteristics, offering hope [1,7]. MBC patients also expressed information needs on end-of-life planning, such as financial planning and burial preferences [4,8].

There is a growing interest in the use of artificial intelligence (AI) within palliative and supportive care. AI has been adopted for use in drug discovery, disease prevention, monitoring, and providing care [9,10]. The usage of AI technology provides patient care beyond in-person consultations. It can also enhance healthcare clinicians’ ability to make informed decisions to improve symptom management, care coordination, and personalized interventions [11]. While technology has shown promise in improving diagnostic accuracy, treatment decision-making, and patient education, few online chatbots specifically address the complex psychosocial and information needs of MBC patients. For example, the app Vik provides general medical information support for breast cancer patients by responding in a text messenger format, addressing topics such as diets, treatment benefits, side effects, and relationship management, with a medication reminder feature for the patients [12]. However, Vik was not specifically designed for MBC patients, whose needs include detailed prognostic information and specific treatment plans. Similarly, Ask Rosa, an emerging app, facilitated digital conversations with breast and ovarian cancer patients about genetic BRCA testing, receiving positive feedback on trustworthiness and usability. However, its evaluation was limited to only two cancer patients [13]. In contrast, there was a formal evaluation of a chatbot called Vivibot, which delivered positive psychology skills and promoted the well-being of 45 young female cancer patients aged 18–29 after their treatment [14]. Compared to the control, the experimental group used the Vivibot for 4 weeks and displayed a trend-level reduction in anxiety with an effect size of 0.41 [14]. Despite its success, Vivibot only focused on emotional support and positive psychology rather than the specific needs of MBC patients. These studies highlight the feasibility of health-focused chatbots and their potential to enhance patient education and support in clinical settings.

To optimize the delivery of cost-effective, person-centred supportive care, we proposed building an automated system to deliver patient education and navigation support to individuals affected by MBC by using curated resources from the Princess Margaret Cancer Centre (PM). These resources, compiled by patient education librarians and experts, currently provide online and in-person educational support for MBC patients and their families. The PM hospital has curated approximately 70 online resources and web links that cover a variety of topics on breast cancer management: physical and psychological symptoms, diagnostics, and treatment options, such as the costs and benefits of surgery, hormonal therapies, biotherapy, and chemotherapy. Additional topics cover family caregiving support for dealing with challenges such as loss and bereavement. The body of information for advanced-disease patients focuses on metastatic diseases and diagnosis understanding, symptom management, such as pain, constipation, diarrhea, anxiety, and depression, and end-of-life discussions such as advanced care planning. The PM also offers e-learning classes available for cancer patients globally. Since the pandemic, there have been over 1300 users from 96 countries enrolled in the University Health Network (UHN) e-learning classes. These classes provide information on specific symptoms, treatment options, treatment side effects, community resources, and lifestyle guidance, such as food safety and exercise.

The objective of this study was to evaluate the extent to which an AI-based chatbot could impact patient activation, focusing on its potential to improve engagement and empowerment among MBC patients. The major advantage of using chatbots is to provide an easy-to-use, dynamic source of education of the basic information about MBC through text-based interactions that simulate human conversation via computer or mobile devices. Patients can ask questions in natural language, and the chatbot delivers information to support informed decision-making and navigation of cancer care resources. The chatbot also explains common physical and emotional symptoms associated with MBC. Evidence suggests that chatbot users exhibit higher adherence to treatment, greater therapeutic alliance, and improvements in psychological well-being and perceived stress compared to non-users. The anonymity of chatbot interactions allows users to disclose sensitive information that might not be shared in in-person interactions [15].

## 2. Materials and Methods

We developed the Artificial Intelligence Patient Librarian (AIPL), a chatbot with software designed to interact with users using decision maps and machine learning algorithms, without human backend intervention. AIPL’s core functions include (1) delivering conversational-based information and advice, (2) inviting users to ask questions on topics of their interest, powered by a deep learning-based natural language processing algorithm, and (3) recommending resources based on the users’ input questions in natural language. The study was conducted in three phases. In Phase 1, the team collaborated with MBC patients and a user-centred designer to translate existing navigation content for delivery by a chatbot. In Phase 2, the team annotated the web resources recommended by patient librarians to generate a list of keywords as inputs to drive the recommender system. A web scraper was used to enrich and re-train the algorithm with updated content. In Phase 3, AIPL was tested with 42 MBC patients to evaluate its functionality and user experience.

The Chatbot Conversation Design: A patient education schematic flowchart was developed with information approved by oncologists (D.C. and M.N.) to teach patients topics including but not limited to disease management, psychosocial and supportive care, and healthcare navigation. The user interface was enhanced with input from oncologists (D.C. and M.N.) and refined with plain language editing and graphics informed by patient education experts (J.P. and T.P.). The conversations in AIPL were clinician-informed through consultations with three oncologists (D.C., M.N., and P.B.) regarding what to include in the chatbot. In Appendix A, Figure A1. represents a list of topics the chatbot rendered for users. The chatbot provided interactive, personalized, and oncologist-approved information to help patients better understand and manage their condition. By responding to user selection and natural language inputs, AIPL tailored its educational content to meet individual patient needs, delivering an engaging and effective learning experience.

Backend Technologies: The development of AIPL incorporated technologies, including ReactJS and a Convolutional Neural Network (CNN) deep learning algorithm to create an interactive chatbot. A web scraper, implemented using BeautifulSoup in Python 3.11, was used to enhance the website resource training dataset. In Appendix A, Figure A2. represents the interactive components of the web application that have been enhanced using HTML, CSS, and JavaScript on the front end. The chatbot was built using the react-chatbot-kit 2.1.0, importing dependencies and creating necessary files including config.js, ActionProvider.js, and MessageParser.js to initialize the chatbot.

Recruitment Procedure: Recruitment for the study began in 2021 and concluded in Summer 2024. Oncologists facilitated the recruitment process by screening the daily breast cancer clinic schedules to identify eligible patients based on their breast cancer subtype. Once potential participants were identified, the attending physician introduced the study to patients during consultations. Interested patients were then approached by a recruiter (A.S. and J.S.) to obtain contact information and further explain the overview of the study. After the appointment, the recruiter sent a consent form to the patient to sign, confirming participation in the study. The participant completed the study in three parts: Part 1 involved completion of a pre-survey, Part 2 the usage and testing of AIPL for a minimum of two weeks, and Part 3, completion of a post-survey. Follow-up emails were sent at each stage to remind participants and ensure participant engagement. After the post-survey, participants received a gift card in appreciation for their participation and an invitation to join a focus group. This study was approved by the institutional UHN Research Ethics Board Ref. #21-5140.

Participants: The study recruited English-speaking female patients aged 18 and older who had been diagnosed with HR+/HER2− advanced breast cancer. Recruitment took place at the M. Lau Breast Center at Princess Margaret Cancer Centre in Toronto, Canada.

Measures: The primary outcome was the Patient Activation Measure (PAM), which provides insights into levels of patient engagement, referring to an individual’s knowledge, skills, and confidence in managing their health to achieve optimal outcomes among populations managing chronic or complex conditions [16,17]. The measure has been shown to be reliable and applicable to cancer patients [17]. Participants completed the PAM-22 survey both before and after using AIPL, reflecting on their cancer experience, treatment, and self-management knowledge [16]. The pre-survey also included the Impact of Event Scale (IES-R) and the Functional Assessment of Cancer Therapy—General (FACT-B) to assess quality of life [14,18]. The post-survey had an additional section evaluating user experience with AIPL using the System Usability Scale (SUS) [19].

Qualitative Measure: To achieve breadth and depth of data from participant’s experience with using AIPL, participants were encouraged to freely express their feelings and thoughts during a semi-structured focus group interview. Eleven participants were invited via email by A.S., with ten agreeing to participate; one could not attend due to time constraints. Focus group sessions were moderated by Y.W.L., while J.S. and A.S. were responsible for taking notes. The meetings, conducted over video conference, each lasted approximately one hour. The discussion questions centred on participants’ experiences using AIPL, its effectiveness in meeting their needs, and suggestions for improvements.

Data Analysis: Descriptive statistics from the pre- and post-surveys were generated using R-Studio. Paired t-tests were conducted to observe changes in responses across both time points. A bi-directional graph was used to visualize the results. Additionally, the McNemar test was applied to each survey question to compare proportions of “Agree” and “Disagree”, assessing changes after using the AIPL. The focus group data were analyzed, identifying the barriers to addressing the medical and psychosocial needs of the MBC patients, as well as the strengths and limitations of the AIPL’s design. We adopted the Reflexive Thematic Analysis method, an inductive approach, allowing codes and themes to emerge from the data. This method emphasizes the researcher’s critical reflection on their assumptions and interpretations, drawing general conclusions from specific observations or data [20].

## 3. Results

### 3.1. Quantitative Findings

Of the 60 participants approached, 18 (30%) declined participation, and 42 (70%) participants consented. Among these, 36 participants (85.7%) completed the study. We asked all completers to participate in the focus group and 10 (27.8%) participated. The majority of the participants were aged 40–64 years. The pre-survey Impact of Event Scale—Revised (IES-R) scores showed a mean of 25.69 (SD = 16.75). The general well-being score from the Functional Assessment of Cancer Therapy—General (FACT-G) was 68.67. The post-survey System Usability Scale (SUS) yielded a score of 72.90 (SD = 14.3) (Table 1).

There were no significant differences in raw Patient Activation Measure (PAM) scores between the pre-survey (mean = 59.33, SD = 5.19) and post-survey (mean = 59.22, SD = 6.16) data. Normalized PAM scores categorized participants as Level 4, indicating they were proactive and highly engaged in their health and followed medical guidance. We investigated changes in responses to individual PAM questions. The responses were aggregated into “Agree”, “Disagree”, and “Not Applicable” categories for better visualization. A non-parametric comparison using McNemar’s test showed no significant differences in the portions of “Agree” and “Disagree” between pre- and post-surveys (Figure 1).

### 3.2. Qualitative Findings

From the experiences of MBC patients who utilized the AIPL, four key themes emerged to describe the extent to which the technology impacted their transition to the advanced stages of the disease. These themes reflect patients’ feedback on the chatbot’s functionality and limitations: (1) AIPL offers basic guidance on wellness and health, (2) AIPL provides limited support for managing relationships, (3) AIPL offers limited medical information unique to their conditions, and (4) AIPL is unable to offer hope to patients. While the findings reveal that MBC patients were generally well-informed experts about their condition, their needs extended beyond basic informational support to focus on emotional well-being and social network support. Figure 2 provides a visual representation of these themes and subthemes, which are explored in detail below.

#### 3.2.1. AIPL Offers Basic Guidance on Wellness and Health

Participants frequently described the AIPL chatbot as an informative and credible source for accessing basic disease information. One participant said “…it had good information, trustworthy information too” (Participant ID 29). They valued the trustworthiness of the vetted content, particularly compared to unreliable online resources. AIPL offers the freedom of choice for patients to pick and choose their topic of interest.

*“What I think is the best thing is that it is vetted information. Nowadays on the web, you can find anything and you don’t know if it’s a reliable source. So coming from you, you know that it’s already been reviewed and that it is correct and scientific and that it’s approved and current information…like I think the thing with the chatbot is like, you choose to investigate that topic, so it’s not like it’s forced up on you”* (Participant ID 24).

Participants also noted the chatbot’s utility and the ease of finding information.

*“I was looking at stats a lot when I was first diagnosed and wanted to know the prognosis and how common this is and that…To have something like this is a great resource, to look things up when you need to…I would use it more when first diagnosed than now just because a lot of the stuff that I ran into I already know…I was diagnosed 8 years ago…”* (Participant ID 35).

Beyond basic information, the chatbot supported various aspects of health maintenance, including emotional health, spiritual care, exercise and diet, and disease management. For example, participants highlighted its guidance on managing sleep issues.

*“I remember thinking I was happy…I think I mentioned to you that one of the categories I was really happy to see was the one on spiritual care…just being able to talk about things…When I was in the hospital, I had spiritual people visit me a couple of times and it was just very soothing. I ran into one of them last week, actually Princess Margaret…and it was just great to reconnect with her…Anyway, anything like that, spiritual, and it doesn’t have to be religious, but just the idea of talking about things on a broader perspective is very helpful”* (Participant ID 35).

*“I was also searching, I’ve been having like serious sleep issues. So I typed in some insomnia and it gave me some information and then said, cognitive behavioural therapy, there’s a manual click here for the manual using the link”* (Participant ID 34).

One patient suggested the chatbot could expand on the topics of intimacy and sexual health:

*“…expanding on some of the questions…intimacy and sex after breast cancer…for a lot of women that are put into chemically induced menopause and also vaginal atrophy and decreased libido. Generally speaking, we have treatment that is hormone-based, lubricant, or even taking hormones, and many of us are not able to do that…I know this is a really really big hot topic and also about body comfort too”* (Participant ID 24).

#### 3.2.2. AIPL Provides Limited Support for Managing Relationships

This theme captures the challenges MBC patients face in managing relationships with loved ones and family members due to the emotional concerns of leaving them behind. While AIPL was useful in gathering basic information, participants felt it fell short in addressing the emotional complexities of maintaining and nurturing relationships. AIPL was able to provide information on organizations for patients to connect with other patients.

*“The reference to organizations like Wellspring, I was really happy to see because they really were a lifesaver to me when I was just diagnosed”* (Participant ID 35).

Participants expressed a desire for AIPL to provide resources that could help them navigate complex relational dynamics. For instance, one patient highlighted the need for tools like support group directories and online forums.

*“If the chatbot can provide a list of lesser-known support groups or if a chatbot could offer me transcripts or online forum…”* (Participant ID 29).

Another participant reflected on the emotional strain relationships can bring, especially when family members struggle to cope.

*“…how to handle people in your life that are dealing with you, caregivers and friends…they’re so upset still, but my mother is just flipped out of the smallest little thing. So I’m not telling her. More support in that area…”* (Participant ID 35).

Patients also discussed their efforts to establish relationships with their healthcare providers, seeking meaningful communication. Some patients shared personal experiences of receiving social and emotional support, emphasizing the importance of personal contact. AIPL was seen as inadequate in fostering these connections. One participant described the value of in-person support groups, such as Wellspring.

*“And for years afterward, I went to a relaxation session every Friday for about 5–6 years. And then they closed the downtown location…It’s a bit of a pain to get there, but the woman who runs the session, she’s just amazing. I can’t say enough about her, so I would follow her anywhere”* (Participant ID 35).

Others saw the potential of AIPL in preparing the patient for consultations with their providers, particularly by helping formulate questions and giving them a clear idea of what to ask. This would be helpful to patients wanting to build a relationship with their provider and primary oncologist.

*“Obviously you should talk to your doctor, but not everybody wants to talk to the doctor first…they don’t know how to initiate it…So they might want the information first because they don’t want to talk to the doctor until they at least have specific questions or made a choice”* (Participant ID 24).

Some patients felt limitations in their relationships with their doctors and wanted to raise specific concerns or questions about alternative treatments. It seemed the doctor was unable to provide alternative approaches to alleviate their disease.

*“Here’s an example: When I go to my oncologist and I say, what can I do? He says nothing. This is the drug you are taking. Then I go to a naturopath or dietitian and ask what can I do? Okay, you can take this food. You can try this. And that is licensed as well. So between nothing and doing something, we all prefer to do something because we are in control in this case”* (Participant ID 11).

Respondents also expressed a strong desire for connections with others who shared the same lived experiences, as this provided comfort and clarity. AIPL was unable to facilitate peer support and could not fulfill this need for the respondent.

*“But for me…I would want to see a chat where people can, If I want to chat and say, ‘Has anyone experience example of sleep insomnia’, and then [someone] comes up and say ‘ohh yes’ and then I can talk to someone who might have the same experience as I and in the same illness…as right now I’m single at home”* (Participant ID 59).

*“I think in the chatbot there could be something about some anecdotal. Could people say their experiences like that? So people can have a frame of reference. You wonder what other people are experiencing? How are they dealing with their thing? So I started going to chat groups…that’s how I learned, what everyone else is dealing with that helped me…That’s what people are looking for when they’re first diagnosed, a frame of reference…What’s normal, what’s not?…What can I expect?”* (Participant ID 35).

#### 3.2.3. AIPL Offers Limited Medical Information Unique to Patient Conditions

This theme highlights how MBC patients, often considered themselves to be “expert patients”, seeking personalized and advanced medical information beyond what AIPL offers. By extensively researching their condition, these patients gain a deep understanding of treatment options, clinical trials, and advancements in cancer care.

This intense focus on learning about their condition serves as both a coping mechanism and a means of empowerment. Patients aimed to actively participate in their care decisions and regain a sense of control over their lives. This all-consuming quest for information can become a central part of their daily routine, offering a sense of purpose and hope in the face of uncertainty. Participants emphasized that AIPL did not adequately address their need for specific, individualized medical information, rarely offering new insights for those already immersed in research. One participant noted the following:

*“The chatbot has a lot of information, but honestly, people like me that like to research, we know it because you go to the sites and to all the resources that we have in this journey. So it was nothing new that I learned…I didn’t find it especially useful because I’ve already knew most of what it answered. It didn’t go beyond what I know…”* (Participant ID 45).

Similarly, another respondent shared their extensive use of online resources and support groups to learn about MBC: “I was diagnosed in July of last year. So a lot of it is new to me, but I’m on some Facebook groups. So I have been learning all the varieties of issues that are occurring and the things are endless that you can look up online” (Participant ID 29).

Despite their extensive knowledge, participants emphasized the difficulty in finding information tailored to their unique medical circumstances. One patient described feeling isolated due to their rare diagnosis:

*“I find I’m a little bit of an outlier everywhere because my situation seems different from everybody else’s…I was told only 3% of the people have a de novo diagnosis, and I have been sent all kinds of possible studies and I never qualify because I’ve not had these other treatments…I’m a bit relieved I didn’t have these very harsh treatments”* (Participant ID 29).

Another participant suggested enhancements to AIPL that could address this gap: “it would be helpful if like when the question is asked ’did I find information useful’, either a live person or maybe an email goes out so that I can get the information I’m looking for” (Participant ID 23).

#### 3.2.4. AIPL Is Unable to Bring Hope or Excitement to Patients

For many respondents, hope and joy came in the form of updates from their oncologists about new drug developments. Patients noted that their optimism and motivation often stemmed from learning about clinical trials, new drug developments, and practical lifestyle guidance. They expressed frustration with AIPL’s inability to provide timely updates or address specific lifestyle concerns. This diminished its perceived value in fostering hope and supporting a better quality of life.

*“I guess what I search for and couldn’t find is anything about clinical trials…I’d like to know what’s next? What’s available…is it something that’s rolled out already FDA approved? Is it something that’s still in trial? What are those trails? How do you know what kind? How effective are those?…I think it’s important to give people with metastatic cancer a little bit of hope that there are the next generation of drugs that are coming out that could be helpful”* (Participant ID 34).

Another participant highlighted the excitement generated by learning about new treatments directly from their doctor:

*“All of a sudden there is a brand new treatment that’s just been approved for my cancer. Luckily, my doctor told me but I might have found out earlier with the chatbot, which is really exciting news because it’s proving to be really effective—just the last two times she [the doctor] has told me about new studies and new trials and they’re really relevant to me because they are about drugs that I’m on…”* (Participant ID 25).

*“But then maybe we can get that type of drug, it’s for that mutation, for that type of cancer. That is the side effects, it can be the progression, survival, whatever those kinds of things will be very helpful because this is what we are expecting, unfortunately, we are living with this disease. what we want is more treatments to live…the more we know and find out about what is available out there, it’s better you know?”* (Participant ID 45).

Beyond medical updates, patients hoped for guidance on returning to normal life despite their diagnosis. Whether planning travel, resuming work, or managing relationships, these areas remained unaddressed by AIPL.

*“And for me having a normal life before you know, I was diagnosed a year ago and I wanted to travel. That was in my cards. I want to travel. I don’t know what I can do and so just ordinary lifestyle topics…I don’t have kids, but if you have children, what are the things you need to…How do you talk to your kids about this? Can you travel safely? Travel tips for example. So I do my own research, but if you’re looking for something that’s a one stop then maybe adding these topics”* (Participant ID 34).

While undergoing treatments and living with MBC, faced with emotional challenges, some respondents hoped for a normal life without having to cope with the disease. The impact of the disease has also changed the patients’ outlook on life, shifting their priorities and responses to everyday experiences. Participants emphasized the emotional toll and the importance of tools that could alleviate anxiety.

*“On one level, little things don’t bother me anymore, like being stuck in a lineup. Little things that people experience on a daily basis, nothing like that bothers me at all now. But if something goes wrong, even the slightest problem (related to the disease)…I’m just in a panic…it’s crazy. I think anything to do with anxiety and alleviating anxiety is just wonderful”* (Participant ID 35).

Patients often turned to alternative treatments when dissatisfied with the limitations of conventional medicine. Certain modalities could not be professionally endorsed by their doctors due to legality issues. AIPL struggled to address this need due to legal and ethical constraints.

*“And I think what I love about your idea is that some of that is just basic things for a day-to-day living perspective…they are practical considerations…if I’m doing something that isn’t good for me, I would love to know that and if somebody asked me why…then provide me with an alternative that is good for me…”* (Participant ID 25).

## 4. Discussion

The current study aims to understand the potential benefits of an AI application for patients with MBC. At baseline, the IES-R responses indicated mild distress, a clinical concern equivalent, suggesting partial Post-Traumatic Stress Disorder (PTSD) symptoms. FACT-G scores also reflected an average quality of life. Quantitative findings showed that AIPL did not significantly impact patient activation, potentially due to participants’ high baseline levels of engagement and self-management. This aligns with prior research showing that women with breast cancer tend to be information seekers [21,22]. However, AIPL demonstrated good usability, as evidenced by the SUS.

The qualitative findings identify four themes from the focus groups which describe the role of AIPL in the MBC patient experience: (1) AIPL offers basic guidance on wellness and health, (2) AIPL offers limited help in providing information unique to their condition, (3) AIPL provides limited support for managing relationships, and (4) AIPL is unable to offer hope to patients.

The first theme suggests that AIPL may be more beneficial for MBC patients in the early stages of breast cancer who need foundational information. MBC patients, who were often well informed and self-identified as “expert patients”, found basic guidance less relevant. However, they valued the credibility of consolidated resources and expressed interest in more personalized medical information, such as insights based on genetic data, to support navigation of the final stages of their disease. The second theme highlights that AIPL is limited in addressing the specific, unique needs of individual patients’ conditions. Respondents noted that the chatbot could not account for rare diagnoses or personalized treatment paths. The third theme reflects challenges in managing relationships with loved ones and navigating complex emotional dynamics, including planning for the future and coping with the impact of the disease on family members. Patients expressed a desire for greater autonomy in treatment decisions and for platforms to share their experiences and connect with others in similar situations. Some patients suggested that AIPL could facilitate peer support through storytelling and discussion forums. The fourth theme underscores the critical role of hope in patients’ psychological well-being. Patients actively sought information about alternative treatments and clinical trials, eagerly pursuing any potential opportunities for improvement. When potential new trial options emerged, they experienced a significant emotional uplift. This finding aligns with previous research emphasizing the importance of hope for individuals facing terminal conditions [1,23].

We found that the processes of learning and self-education seemed to keep patients hopeful and, in some cases, enabled them to embrace a new identity as “expert patients”. The quantitative finding, using the PAM, also supported this qualitative finding of patients being proactive with their health, highly engaged, and adhering to medical guidance. This suggests that patient education not only provides empowerment and practical knowledge but also improves patient emotional well-being. Establishing in-person study groups could further support patients by fostering social interaction and enabling the exchange of lived experience, which may help them navigate personal relationships more effectively. Unfortunately, due to current legal and ethical constraints in healthcare, including the European AI Act [24], AIPL cannot provide highly specific information about personalized new treatments. The principle of “human-in-the-loop” necessitates oversight by trained medical professionals to deliver high-stakes information. Nevertheless, AIPL has the potential to assist patients who want a return to how life was or to find a new normal. AIPL offers practical guidance, such as assisting patients in planning vacations or returning to work, allowing them to live with MBC without letting it dominate their lives.

AIPL addresses some gaps in the literature. In particular, AIPL is useful for patients who are English-speaking and comfortable with technology use. Previous interventions have met the psychological and informational needs of MBC patients through in-person and telephone sessions. There are a few chatbot interventions, including the My Alma app, ASCAPE framework, Ask Rosa, and Vik, but no patient outcomes have been reported. Therefore, their impact on important outcomes such as the ability to self-manage remains unclear [12,13,25,26]. This study provides valuable insights into how AI can enhance patient care and emotional well-being, highlighting areas for future development in AIPL’s functionality and patient support.

AIPL shares similarities with Vivibot, which has undergone comprehensive evaluation. While Vivibot caters to general cancer patients, AIPL addresses the support needed for MBC patients, as they typically require more specific treatment plans and face poorer prognoses compared to patients with early-stage breast cancer. Vivibot was designed for young adults below 40 years of age, whereas AIPL was tested predominantly by patients aged 40–64 years old. Both interventions demonstrate the potential of technology to enhance mental well-being and engagement in cancer care [8]. Participants in both studies valued the convenience of having a supportive tool available at any time, allowing them to interact with content at their own pace and encourage continued use. Vivibot’s qualitative feedback highlights its non-judgemental nature and the impact of relatable videos and personal stories, features highly rated by young women in the study [14]. However, unlike Vivibot, which operates on Facebook Messenger, raising privacy concerns due to the platform’s history of cybersecurity issues, AIPL was deployed on a secure server at the University Health Network, a public research and teaching hospital network, in Toronto, Canada, with no tracking of user activities. Moreover, our study focused on MBC patients on life-sustaining treatments, whose goal was to continuously explore new treatment possibilities. AIPL was designed to provide oncologist-approved medical resources, guidance on symptom management, end-of-life decision support, and advanced treatment options, addressing a critical gap in the literature.

The public release of ChatGPT in 2022 during the study’s recruitment period significantly influenced participants’ expectations of AIPL, as they compared it to ChatGPT’s more seamless and intelligent conversational capabilities. Many patients found AIPL less intuitive and intelligent in comparison, leading to suggestions for improvements, like a back-and-forth seamless dialogue response experience. In view of this, AIPL has been updated to integrate Phi-3, a large language model designed to provide more advanced and responsive support tailored to MBC patients. However, the study has limitations. AIPL was tested at the PM, a setting that predominantly serves an affluent and highly educated population, limiting the generalizability of findings to more diverse demographics. Future research should explore the use of AIPL in broader and in early-stage breast cancer patient populations to enhance its applicability and impact.

Future work should include testing an enhanced version of the AIPL (https://aipl.uhn.ca) which builds on insights from this study and others to better address patient informational needs and preferences. This upgraded chatbot aims to provide more comprehensive and personalized support by incorporating generative AI technologies. These advancements could help educate patients about the latest treatments tailored to their medical histories and offer personalized guidance on understanding scientific studies and interpreting medical findings. In particular, a digital MBC patient will be built featuring a warm persona that shares lived experiences and answer users’ lifestyle questions, including relationship and intimacy concerns, to support patients emotionally. While the full extent of the latest technology’s ability to address all patient needs is still being evaluated, AIPL demonstrates the idea of leveraging AI to improve care and support for advanced breast cancer patients. Future iterations of AIPL could also serve as a learning platform, connecting patients beyond study groups to encourage shared knowledge and experiences.

## 5. Conclusions

This study has provided valuable insights into the needs and preferences of MBC patients, which have informed the development of a more advanced version of AIPL. While advanced digital healthcare tools are essential, they must be rooted in a patient-centred approach that addresses the individualized and multifaceted needs of patients. We remain hopeful that ongoing technological advancements will enable the integration of various aspects of care to ensure that patients receive holistic and individualized support.

## Figures and Tables

**Figure 1 curroncol-32-00145-f001:**
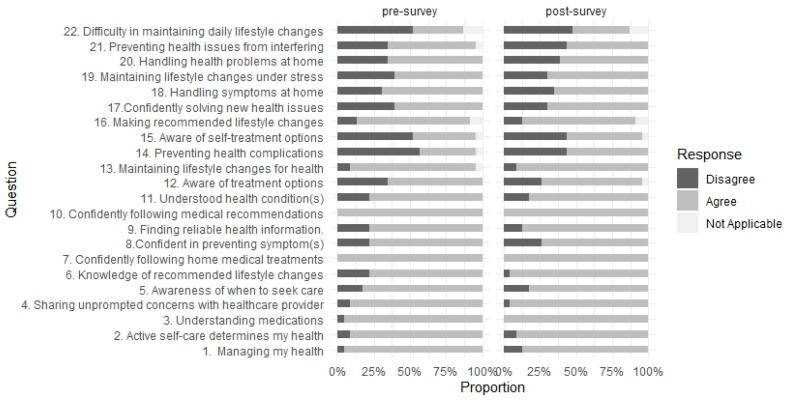
Frequency of response of the Patient Activation Measure (PAM).

**Figure 2 curroncol-32-00145-f002:**
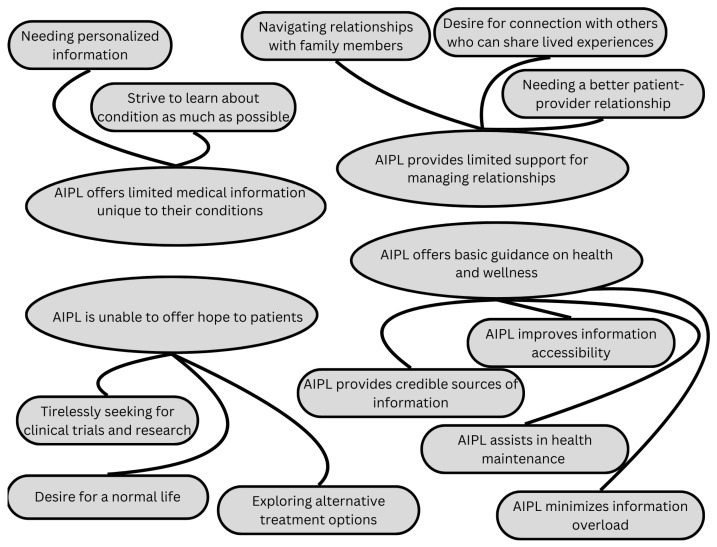
Summary of themes and subthemes.

**Table 1 curroncol-32-00145-t001:** Baseline patient characteristics (n = 42).

Characteristics		
**Age range (years)**	**n**	**%**
18–39	3	7.1
40–64	26	61.9
65+	2	4.8
Do not want to disclose	11	26.1
	**Mean**	**SD**
**Patient Activation Measure (PAM) raw score**	59.33	5.19
**Normalized PAM score**	78.49	16.74
**Pre-survey IES-R**	25.69	16.75
**PRE-survey FACT-B subscales**		
Physical well-being (PWB)	20.22	4.81
Social/family well-being (SWB)	17.97	5.55
Emotional well-being (EWB)	14.30	4.13
Functional well-being (FWB)	16.22	5.86
**Post-survey System Usability Scale (SUS)**	72.90	14.3

## Data Availability

The data presented in this study are available on request from the corresponding author, Yvonne Leung.

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
