# Peer review of "The Extent to Which Artificial Intelligence Can Help Fulfill Metastatic Breast Cancer Patient Healthcare Needs: A Mixed-Methods Study"

_curroncol, 2025, doi:10.3390/curroncol32030145_

Round 1

Reviewer 1 Report

Comments and Suggestions for Authors

The article discusses the use of a chat bot to develop an educational resource for patients with metastatic breast cancer - the study is clearly and thoughtfully developed and presented and the limitations of the chat bot with respect to providing hope are well discussed in particular. The outcomes are well presented in text and in figures - the patient testimonies are particularly well integrated to support the conclusions. The advent of chat gpt leap frogged the technology used in the study but the messages evident in the paper are applicable to all 

References including regulatory documents are comprehensive 

My only comment is that the abstract is long and could be shortened to increase readability 

Reviewer 2 Report

Comments and Suggestions for Authors

Authors should clarify at the title that it is a qualitative study. It is not a mixed method.

Results section should have 2 subsections : 1. the quantitative results and 2. the qualitative results.

References int the text. Why thenumbers of references in the introduction start with the numbers 11, 23 etc. The references shouls start with 1, 2, 3 etc

Comments on the Quality of English Language

Major english editing is needed

Reviewer 3 Report

Comments and Suggestions for Authors

Regarding the manuscript titled "The extent to which artificial intelligence can help fulfill metastatic breast cancer patient healthcare needs: a mixed-methods study"
I would like to inform the authors that this manuscript is scientifically valuable. Its methodology is well described and innovative. However, it still seems to need limited improvements to improve the quality. To improve the quality of this study, I recommend that you pay attention to the following points.
1: Enrich the abstract. Try to include all the study findings in the abstract.
2: Your manuscript suffers from one issue, which is the lack of a basic definition and description of artificial intelligence and its applications in the field of healthcare. This section will greatly improve the readability of your manuscript. You can add a half paragraph to this section. You can also use the following resources for this purpose.
"Artificial intelligence in drug discovery and development against antimicrobial resistance: A narrative review"

"Mobile apps for COVID-19 detection and diagnosis for future pandemic control: Multidimensional systematic review"

These sources can be good sources for this paragraph and. On the other hand, they can enrich your reference list.

3: Present the results section with more and better charts and images. This section has good examples, but its presentation needs more charts and images.

The rest is well presented and by making these changes, the quality of the manuscript can be greatly improved.
